# Functional Characterization of Tomato ShROP7 in Regulating Resistance against *Oidium neolycopersici*

**DOI:** 10.3390/ijms23158557

**Published:** 2022-08-02

**Authors:** Yanan Meng, Ancheng Zhang, Qing Ma, Lianxi Xing

**Affiliations:** 1College of Life Sciences, Northwest University, Xi’an 710069, China; mengyanan@nwu.edu.cn; 2College of Plant Protection, Northwest A&F University, Xianyang 712100, Chinamaqing@nwsuaf.edu.cn (Q.M.); 3Key Laboratory of Resource Biology and Biotechnology in Western China, Ministry of Education, Northwest University, Xi’an 710069, China

**Keywords:** tomato, *Oidium neolycopersici*, plant resistance, rho of plants (ROPs)

## Abstract

ROPs (Rho-like GTPases from plants) are a unique family of small GTP-binding proteins in plants and play vital roles in numerous cellular processes, including growth and development, abiotic stress signaling, and plant defense. In the case of the latter, the role of ROPs as response regulators to obligate parasitism remains largely enigmatic. Herein, we isolated and identified *ShROP*7 and show that it plays a critical role in plant immune response to pathogen infection. Real-time quantitative PCR analysis revealed that the expression of *ShROP7* was significantly increased during incompatible interactions. To establish its requirement for resistance, we demonstrate that virus-induced gene silencing (VIGS) of *ShROP7* resulted in increased susceptibility of tomato to *Oidium neolycopersici* (*On*) Lanzhou strain (*On*-Lz). Downstream resistance signaling through H_2_O_2_ and the induction of the hypersensitive response (HR) in *ShROP7*-silenced plants were significantly reduced after inoculating with *On*-Lz. Taken together, with the identification of ShROP7-interacting candidates, including ShSOBIR1, we demonstrate that ShROP7 plays a positive regulatory role in tomato powdery mildew resistance.

## 1. Introduction

The small GTP-binding proteins constitute a superfamily of signaling regulators, among which are the Ras, Rho, Rab, Arf/Sar, and Ran subfamilies in yeast and animals, and the Rho, Rab, Arf/Sar, and Ran subfamilies in plants [1]. The Rho-related subfamily of plant small monomeric GTPases is referred to as RAC or ROPs (Rho-like GTPases from plants) [2,3]. As indicated by their nomenclature, this family of small proteins with molecular weights in the range of 20–40 kDa, and possesses a GTP binding active site with concomitant GTPase activity [2,4]. In plants, small GTP-binding proteins are mainly involved in plant signal transduction processes, with critical roles in controlling cell polarity, growth, morphological development, cytokinesis, hormone signaling, control of cytoskeletal organization, and response to pathogen infection [5,6,7,8,9].

As important signal transduction regulators in plants, ROPs play vital roles in plant defense, regulating defense and immune-related processes associated with biotrophic, necrotrophic, and obligate parasites [6,7,8]. Upon pathogen penetration of the host cell wall, immune signal transduction is initiated via the activity of receptor-like kinases (RLKs), pattern recognition receptors (PRRs), which perceive initial signals associated with the invading pathogen, ultimately transducing these signals to the guanine nucleotide exchange factors (GEFs) [10,11,12,13]. Once initiated, signaling transfers to the ROP-GEF signaling node, resulting in the conjugation and conversion of the GDP-bound inactive (GDP·ROP) into GTP-bound active (GTP·ROP). The key function of this process is the transmission of extracellular signaling to the intracellular defense signaling network.

In parallel to the extra-to-intracellular signal transduction cascade, the GTPase-activating protein (GAP) converts the GTP·ROP node to GDP·ROP by promoting GTPase activity. The function of this conversion is to maintain a dynamic balance of the two ROP protein states. This activity is further regulated by the activity of the guanine nucleotide dissociation inhibitor (GDI), which inhibits GDP·ROP to GTP·ROP conversion [6]. The sum of these signaling processes, and the generation of an activated ROP protein, triggers the activation of downstream signals which function to initiate additional, required, stress response cascades (e.g., Ca^2+^, ABA responses, H_2_O_2_, actin cytoskeleton) [14,15].

The function of the ROP family in the interaction between plants and pathogens remains largely enigmatic [16]. ROPs have been shown to possess critical functions which regulate the establishment of plant pathogen-associated molecular pattern (PAMP)-triggered immunity (PTI). In this role, it has been demonstrated that *OsRac1* regulates reactive oxygen species (ROS) production and cell death, while constitutively active (CA) *OsRac1* results in enhanced PAMP-induced ROS production and resistance to *Xanthomonas oryzae* pv. *oryzae* and *Magnaporthe grisea* in rice [17,18]. As a function of effector-triggered immunity (ETI), it has been further shown that *OsRac1* is activated by the R-protein Pit for triggering downstream responses in [10]. This role was further demonstrated by the silencing of a dominant negative (DN) variant of *OsRac1* in tobacco resulting in a reduction of pathogen-induced ROS and hypersensitive response (HR) production [19]. This function was further established via the overexpression of *HvRac1* in barley, which led to a promotion of callose deposition and ROS production, the sum of which provided enhanced resistance to *M. oryzae* penetration [20].

It is not surprising that various members of the ROP family from the same plant play distinct roles in disease resistance. For example, among seven *ROP* family genes from rice (*Oryza sativa*), *OsRac1* has been shown to play a positive regulatory role in resistance to pathogens, while *OsRac4*, *OsRac5*, and *OsRac6* negatively regulate rice resistance to pathogens. Further, *OsRac3* and *OsRac7* have been shown to not be involved in plant resistance [21,22]. A similar functionality has been shown in wheat, whereby *TaRab18* is required for stripe rust resistance [23], yet *TaRac6* and *TaRop10* are susceptibility factors in wheat–*Puccinia striiformis* f. sp. *tritici* interaction, respectively [16]. Similarly, in barley, *HvRacB* promotes susceptibility to *Blumeria graminis* f. sp. *hordei* [1], while CN-*HvRac1* promotes resistance to *M. oryzae* [20].

Tomato powdery mildew, caused by *Oidium neolycopersici* (*On*), is an obligate biotrophic ascomycotina fungal disease that causes tremendous economic losses in tomato (*Solanum lycopersicum* L.). As an obligate biotrophic ascomycotina, *On* can parasitize more than 60 plants, including members of the Solanaceae [24]. At present, due to the warm and humid environmental conditions in greenhouse settings, tomato powdery mildew has become more serious, with the incidence of infection rates being 80–90%—a major problem in tomato production around the world [25]. As such, studies on the mechanism(s) underpinning resistance to powdery mildew in tomato have become a prerequisite for tomato disease-resistance breeding. In the current study, we found that the expression of *ShROP7* was significantly upregulated during an incompatible host-pathogen interaction, with further analyses revealing that the silencing of *ShROP7* enhances the susceptibility of tomato to *Oidium neolycopersici* (*On*) Lanzhou strain (*On*-Lz). As a potential mechanism driving this function, we demonstrate that a receptor-like kinase (RLK) protein ShSOBIR1 interacts with ShROP7. In total, the current work described herein provides evidence that ShROP7 plays a critical role in the resistance of tomato to powdery mildew.

## 2. Results

### 2.1. Identification and Sequence Analysis of Tomato ShROP7

A tomato 594-bp homolog of Rac-like GTP binding protein 13 (Accession number Solyc02g077400.4.1) was isolated from tomato cv. LA1777 by homology-based cloning. The predicted CDS of *ShROP* encodes a protein of 197-amino acid with CaaL motif. Phylogenetic analysis of ShROP with other members of tomato ROPs and *Arabidopsis thaliana* ROPs (i.e., ROP1-11) showed that ShROP has the closest relationship with NaRAC13, with the next closest relationship being AtROP7. Based on the output of this analysis, we named the predicted protein ShROP7 (Figure 1A). The amino acid sequence of ShROP7 was compared with *A. thaliana* AtRAC-like2, *Nicotiana attenuata* NaRAC13, *N. tabacu*m NtRAC13, *Solanum pennellii* SpRAC13, *S. tuberosum* StRAC13, *Cucumis sativus* CsRAC13, *Capsicum annuum* CaRAC13, *Vitis vinifera* VvRAC13, and *Spinacia oleracea* SoRAC13, and the compared results showed that the N-terminus of ShROP7 was highly conserved across the ROP/RAC superfamily (Figure 1B).

### 2.2. ShROP7 Cannot Induce Cell Death nor Suppress BAX-Induced Necrosis

In order to define the role of *ShROP7* during pathogen infection and disease resistance signaling, the PVX-based transient overexpression system in *Nicotiana benthamiana* leaves was used to evaluate *ShROP*7 function in cell death elicitation. Our aim was to evaluate the overexpression cell death activity of *ShROP7* and/or its ability to suppress PCD. *Agrobacterium* harboring pGR106 (negative control), pGR106::*ShROP7*, and buffer (blank control) were infiltrated into tobacco leaves, respectively, and no cell death phenotypes were observed (Figure 2B,C; sites 1, 2 and 6). BAX, a cell death-promoting member of the Bcl-2 family of proteins that triggers cell death when expressed in plants [26], was used as positive control. At 24 hpi, BAX, pGR106::*ShROP7* and pGR106 sites were injected with BAX to evaluate whether ShROP7 can suppress BAX-induced cell death (Figure 2A; sites 3, 4 and 5). At 5 days after being infiltrated with *Agrobacterium*, ShROP7 did not induce cell death, nor could it suppress BAX-triggered cell death (Figure 2B,C).

### 2.3. ShROP7 Is Differentially Induced by On-Lz Infection

To determine the function of *ShROP7* in the tomato-powdery mildew interaction, the mRNA accumulation of *ShROP7* was monitored during the interaction between tomato and *On*-Lz at 0, 12, 18, 24, 36, 48, 72, 96, and 120 hpi. As shown in Figure 3, the expression of *ShROP7* was significantly up-regulated at 18, 24, 72, 96 and 120 hpi during an incompatible reaction, in which tomato cv. LA1777 constitutes an incompatible reaction with *On*-Lz and peaked at 96 hpi (ca. 5.39-fold compared to 0 hpi). In addition, the accumulation of *ShROP7* was significantly reduced at 18, 24, 36 and 48 hpi compared with 0 hpi in a compatible reaction, in which tomato cv. MM constitutes a compatible reaction with *On*-Lz. Therefore, *ShROP7* is differentially induced by *On*-Lz infection.

### 2.4. Silencing of ShROP7 Enhances the Susceptibility of Tomato to On-Lz

Based on that the expression of *ShROP7* increased in incompatible reaction after *On*-Lz infection, a tobacco rattle virus-induced gene silencing (TRV-VIGS)-based approach was used to silence *ShROP7* in tomato cv. LA1777. A 275-bp fragment of *ShROP7* was cloned into pTRV2 vector, and the expression of *ShROP7* was reduced by the TRV-VIGS system to observe the function of *ShROP7*. The TRV2::*ShROP7*, TRV2::*ShPDS* (positive control) and TRV2 (negative control) were infiltrated in the four-leaf LA1777 and MM. Thirty days after inoculating with TRV2::*ShPDS*, the photo-bleaching symptoms were observed on the new leaves (Figure 4B,F), whereas no significant changes were observed on TRV2::*ShROP7* (Figure 4C,G) and CK (Figure 4A,E) plants. Concurrently, the leaves at the same position were sampled in TRV2::*ShROP7* and control plants, respectively. The expression level of *ShROP7* was calculated by RT-qPCR, with the accumulation of *ShROP*7 being reduced by 68.9% (Figure 4D) in TRV2::*ShROP7* plants compared with the control in LA1777 plants and reduced by 44.6% (Figure 4H) in MM plants, which can be used to verify the role of *ShROP7* in the resistance to *On*-Lz.

When TRV2::*ShPDS* plants showed robust signs of photo-bleaching, leaves of *ShROP7*-silenced and control plants were inoculated with *On*-Lz. The infection phenotypes of the tomato leaves were observed, and disease indexes were recorded at 7 and 14 dpi, respectively. In LA1777 silenced plants, as shown in Figure 5A, the leaves of CK and TRV2::*ShROP7* developed reduced areas of powdery mildew disease spot lesions, while the disease indexes of TRV2::*ShROP7* plants were significantly higher than control plants at 7 dpi (Figure 5C). At 14 dpi, obvious powdery mildew disease lesions were produced on the infected leaves, and *ShROP7*-silenced plants produced more disease spot lesions compared with control (Figure 5B), with the disease index of TRV2::*ShROP7* plants being 12.7, which was significantly higher than that of the control plants (Figure 5C). On the contrary, the disease indexes of CK and TRV2::ShROP7 had no significant change in MM plants (Figure 5 D–F).

To determine whether *ShROP7* regulates tomato resistance to powdery mildew through SA or JA signaling pathways, the expression of the SA-dependent disease resistance signal pathway marker gene *ShPR1* and the JA-dependent disease resistance signal pathway marker genes were evaluated. As shown in Figure 6A,B, the expression of *ShPR1* was significantly reduced in *ShROP7*-silenced plants compared with control plants at 24, 48 and 120 hpi. The mRNA accumulation of *ShPDF1.2* had no significant change, indicating that tomato LA1777 is likely transduced through the SA signaling pathway as opposed to the JA pathway, to resist powdery mildew infection.

### 2.5. Histological Observation of LA1777 ShROP7-Silenced Plants

To further define the resistance changes of tomato infected by *On*-Lz after silencing *ShROP7*, the expression levels of *ShCAT* and *ShSOD* were monitored. As shown in Figure 6C,D, the mRNA accumulation of H_2_O_2_ pathway related genes *ShCAT* and *ShSOD* were significantly induced at 48 and 120 hpi. Additionally, we measured the production rate of H_2_O_2_ and hypersensitive response (HR) at 6, 18, 24, 48, and 72 hpi in LA1777 *ShROP7*-silenced and control plants. After DAB staining, and as shown in Figure 7A,B, the production of H_2_O_2_ was not observed at 6 dpi, and a small number of H_2_O_2_ were produced at 18 hpi. At 24, 48, and 72 hpi, the production rate of H_2_O_2_ in TRV2 plants was significantly higher than that of silenced plants (*P* < 0.01) (Figure 7B). In *ShROP7*-silenced plants, the production rate of H_2_O_2_ was increased gradually as the infection progressed, being 7.3%, 8.3% and 10.7% at 24, 48 and 72 hpi, respectively. Similarly, the production rate of H_2_O_2_ increased rapidly in the control plants, which reached 25% at 72 hpi. As a second, parallel, HR, cell death was stained with trypan blue. As shown in Figure 7C,D, the production of HR was not observed at 6 dpi. At 18 hpi, there were no obvious differences between TRV2 and TRV2::*ShROP7* plants in HR production. With the increase of haustoria formation at 24 hpi, the elicitation of the HR was 22.3% in TRV2-silenced plants, which was significantly higher than in *ShROP7-*silenced plants (ca. 7.7%). At 48 hpi, the elicitation of the HR in control plants was approximately 28%, 2.9 times higher than in TRV2::*ShROP7* plants. At 72 hpi, the rate of HR induction in control plants was 37.7%, which was significantly higher than the rate of *ShROP7*-silenced plants (ca. 11%). Taken together, these results indicate that a reduction in *ShROP7* expression compromises resistance signaling in response to *On*-Lz infection.

### 2.6. Screening and Identification of ShROP7 Interaction Proteins

In the current study, tomato cv. LA1777 leaves infected by *On*-Lz were used to construct a yeast two-hybrid cDNA library of tomato-powdery mildew incompatible interaction, with the aim of screening out proteins interacting with ShROP7. The CDS sequence of *ShROP7* was cloned into BD vector as bait and was transformed into the yeast strain Y2H Gold. After testing the bait construct for autoactivation and toxicity, the cDNA library was screened using ShROP7 as a bait. The development of blue yeast colonies on SD/-Leu-Trp-His-Ade+Aba+X-α-Gal medium was used as a selection criterion for putative positive interacting proteins following the yeast protocols handbook (Takara, Dalian, China). Using this approach, we identified 35 candidate proteins that interact with ShROP7. Among the candidate interacting proteins, we identified and confirmed interaction between ShROP7 and SOBIR1, wound-induced proteinase inhibitor 1, glutathione peroxidase (GSHPx), pectin acetylesterase (PAEs) and chitinase 3 related to plant defense response; eukaryotic initiation factor 4A-2, tyrosine-protein kinase 3, serine/threonine-protein kinase sty46 ferredoxin, thioredoxin, MAPK16, E3 ubiquitin protein related to cell signaling (Table 1). Herein, we focused on SOBIR1, a receptor-like kinases (RLKs) protein containing a signal domain, an extracellular leucine rich repeat (LRR) domain, transmembrane domain and serine/threonine kinase domain, which is a key enzyme that interacts with both Cf-4 and Ve1 to require for the Cf-4- and Ve1-mediated HR in tomato and *Arabidopsis thaliana*, respectively [27]. The CDS sequence of *ShSOBIR1* without a signal peptide (25–638 amino acid) was inserted into the yeast two-hybrid AD vector and the reconstituted *ShROP7*+*ShSOBIR1* and negative controls (AD+BD, AD+*ShROP7* and BD+*ShSOBIR1*) were transformed into yeast strain Y2H Gold. As shown in Figure 8A, the transformants were grown on SD/-Trp-Leu+Aba medium, indicating successful co-transformation. In addition, only *ShROP7*+*ShSOBIR1* containing yeast cells developed blue colonies on SD/-Leu-Trp-His-Ade+Aba+X-α-Gal medium, indicating that ShROP7 and ShSOBIR1 can activate the GAL4 yeast expression system.

To further confirm the interaction between ShSOBIR1 and ShROP7, we conducted bimolecular fluorescence complementation (BiFC) to verify the in vivo interaction. The ORFs of *ShROP7* and *ShSOBIR1*, without a signal peptide, were inserted into pSPYCE and pSPYNE, respectively. The recombinant plasmids were transferred to *Agrobacterium* resultant transformants which were infiltrated into tobacco leaves. A 35S-pSPYNE+35S-pSPYCE was used as a negative control. After 48–72 h, *Agrobacterium*-infiltrated cells were evaluated and yellow fluorescence was observed by laser confocal microscopy. As shown in Figure 8B, we observed that ShROP7 and ShSOBIR1 co-localized to the nucleus and on the cell membrane. In short, this result supports the presence of an interaction between ShROP7 and ShSOBIR1 *in planta*.

## 3. Discussion

Small GTP-binding proteins exist ubiquitously in eukaryotes and constitute a superfamily. As molecular switches, they regulate a series of physiological processes including signaling transduction, cell proliferation, composition of cytoskeleton, cell membrane transporting and gene expression [28]. The ROPs subfamily of small GTP-binding proteins transmits a variety of extracellular and intracellular signals, which can amplify specific signals by acting as the main switch in the early stage of signal cascade [2]. In *Arabidopsis*, *AtROP2* and *AtROP11* increase resistance to *Pseudomonas syringae* pv. *tomato* (*Pst*) DC3000 in transgenic overexpression plants. However, *AtROP10* has the opposite effect on resistance to *Pst* [29]. Together, these studies show that ROPs play both positive and negative roles in the establishment of plant defense. At present, there are few reports about ROPs participating in tomato powdery mildew resistance, and the disease resistance mechanisms of ROPs remain largely undefined.

The ROP family contains a small GTP-binding protein domain, containing five highly conserved G-boxes, one of which, the G-box, functions to bind effectors. The other four G-boxes function in binding GTP/GDP and the hydrolysis of GTP to GDP [30]. Functionally, the C-terminus of plant ROPs is highly variable. Related to their activity and function as GTPases, which are divided into type I and II types, type I ROPs terminate with a conserved CaaL motif and undergo prenylation, whereas type II ROPs lack the CaaL motif and undergo S-acylation [31]. Further, type I ROPs are divided into subtypes 1 and 3, while type II ROPs constitute subtype 2. Among the 11 Arabidopsis ROPs, *AtROP7* and *AtROP8* are the subtype 1 of ROPs, subtype 3 includes *AtROP1*- *AtROP6*, and subtype 2 includes *AtROP9*- *AtROP11* [32]. In this study, the protein sequence of ShROP was found to have the closest relationship with NaRAC13, with the next closest relationship being with AtROP7. In addition, based on the presence of a CaaL motif within the C-terminus of the protein, we conclude that ShROP7 belongs to the type I ROP family.

ROP7 is one of the original members of the ROP family, hypothesized to have evolved with the emergence of the plant xylem [33]. Indeed, *AtRAC2/ROP7* has been found to be related to the formation of secondary walls in the primary xylem of plants, and overexpression of *AtROP7* shortens the leaf length, indicating that *AtROP7* plays an important role in the formation of the plant actin cytoskeleton [34]. Additional studies have shown that *AtROP7* is stimulated by phytochrome B, required for the opening and closing movement of stomata [35]. Herein, RT-qPCR results revealed that the expression of *ShROP7* was significantly up-regulated in an incompatible reaction, which was required for tomato resistance to powdery mildew. To support this, we observed that the silencing of *ShROP7* enhanced the susceptibility of tomato cv. LA1777 to *On*-Lz, with disease indexes being significantly higher than those of control plants in tomato cv. LA1777 *ShROP7*-silenced plants at 7 and 14 days after inoculating with *On*-Lz. Taken together, these data support the hypothesis that *ShROP7* positively regulates tomato resistance to powdery mildew.

Among the fastest defense responses activated following pathogen infection is the generation of reactive oxygen species (ROS), a process required for the induction of the HR and the abrogation of pathogen infection and proliferation. Mechanistically, the HR reaction results in an oxidative burst, producing reactive oxygen intermediates (ROIs) that contain hydrogen peroxide (H_2_O_2_), superoxide radical anions (O^−2^) and hydroxyl radical (OH^−^) [36]. Herein, we observed that the production of H_2_O_2_ and HR were both significantly reduced in tomato infected by *On*-Lz after silencing *ShROP7*. Moreover, we observed that the expression of H_2_O_2_ pathway related genes *ShSOD* and *ShCAT* were induced in *ShROP7* plants. These observations suggest that *ShROP7* is required for the regulation of broad defense signaling processes in tomato required resistance to powdery mildew.

The data presented above confirm that the overexpression of *ShROP7* does not induce localized cell death, nor does it inhibit cell death caused by BAX. However, our results do affirm that the silencing of *ShROP7* significantly reduces the generation of H_2_O_2_ and the formation rate of HR, both of which are required for robust immune signaling and defense. To investigate the mechanism(s) through which *ShROP7* regulates the activation of the HR response and ROS accumulation, we sought to identify specific protein–protein interactions with ShROP7 which might regulate these cascades. Through this approach, we identified a specific interaction between ShROP7 and ShSOBIR1, an RLK protein which has been shown to regulate broad host defense signaling processes. For example, the silencing of *SOBIR1* and *SOBIR1*-like genes has been shown to affect pathogen-induced callose deposition, the production of reactive oxygen species, and the expression of PTI marker genes. Further studies have narrowed this function to demonstrate that the kinase domain of SOBIR1 is essential for eliciting the (ParA1)-based induction of SOBIR1 cell death [37]. Additionally, several proteins, such as OsMAPK6, CERK1, GEF1, and SPL11, have been shown to be associated with ROP-based signaling during host–pathogen interactions [38,39,40].

In conclusion, we demonstrate that ShROP7 is a type I ROP that is required for robust immune responses in tomato following infection with the powdery mildew pathogen *On*-Lz. Through its interaction with ShSOBIR1, ShROP7 plays a role in governing the level of tomato resistance to *On*-Lz by affecting the generation rate of H_2_O_2_ and the rate of HR development. Taken together, the data presented herein further our understanding of the role of the ROP family in regulating disease resistance signaling in plants.

## 4. Materials and Methods

### 4.1. Plant and Pathogen Materials and Growth

Tomato cultivar (cv.) Moneymaker (MM) (*Solanum lycopersicum*) and *S. habrochaites* LA1777 were used in this research. *Oidium neolycopersici* (*On*) Lanzhou strain (*On*-Lz) was propagated and preserved according to the method of Sun et al. [41]. The tomato cv. MM and LA1777 were grown at 25 °C, with a day/night cycle of 16 h/8 h. LA1777 is a highly resistant tomato powdery mildew cultivar and constitutes an incompatible reaction with *On*-Lz, while obvious powdery mildew disease lesions appeared in tomato cv. MM and constitute a compatible reaction with *On*-Lz. Tomato seeds were surface sterilized with 3% sodium hypochlorite for 3 min, and then immediately rinsed with distilled water three times [41]. After surface sterilization, seeds were cultivated and germinated in vermiculite seedling trays, and grown for one week under conditions of 25 ± 1 °C with 90% relative humidity. After the emergence of the cotyledon, seedlings were transplanted into 15 cm pots.

Tobacco (*Nicotiana benthamiana*) was grown at 25 °C under a 14 h light/10 h dark cycle. After 4–6 weeks, tobacco leaves were inoculated with *Agrobacterium tumefaciens* strain GV3101 expressing pGR106*-*gene, pSPYCE-gene, and pSPYNE-gene vectors for protein overexpression or Bimolecular fluorescence complementary (BiFC) analysis, respectively.

*Escherichia coli* strain DH5α was grown on Luria–Bertain (LB) solid medium at 37 °C. *Agrobacterium tumefaciens* strain GV3101 was grown on LB media containing rifampicin and gentamicin at 28 °C. *Saccharomyces cerevisiae* strain Y2H Gold with four reporter genes (AbAr, HIS3, ADE2 and MEL1) was grown on YPDA (1% yeast extract, 2% peptone, 2% dextrose, 0.02% adenine) medium at 37 °C.

### 4.2. Identification and Sequence Analysis of ShROP

Tomato Rac-like GTP binding protein-13 (Solyc02g077400.4.1) amino acid sequence was used as query in the SGN Tomato Combined database (https://solgenomics.net/tools/blast/, accessed on 9 December 2019), the BLASTP program against an Arabidopsis protein database (http://www.arabidopsis.org/Blast/index.jsp, accessed on 20 December 2019) and the NCBI BLASTN of GenBank (https://blast.ncbi.nlm.nih.gov/Blast.cgi, accessed on 20 December 2019) to search for homologous sequences. DNAMAN 6.0 software (Lynnon BioSoft) was used for sequence comparisons with default settings. Phylogenetic analysis was performed with the MEGA 6.0 software package (http://www.megasoftware.net/, accessed on 5 January 2020) using the proximity method. Protein domains were predicted using the SMART (http://smart.embl-heidelberg.de/, accessed on 10 January 2020) and UniProt protein databases (https://www. uniprot.org, accessed on 10 January 2020).

### 4.3. Agrobacterium-Mediated Transient Overexpression in Tobacco (Nicotiana Benthamiana)

For transient overexpression in the tobacco, DNA primers (Appendix A) were used to clone the full-length *ShROP7* cDNA via *Sal*I digestion and ligation. The intelligent seamless cloning kit (Takara, Japan) was used to link the target gene to pGR106. The recombinant plasmids pGR106::*ShROP7* and pGR106 (negative control) were transformed into the *A. tumefaciens* strain GV3101. For *A. tumefaciens*-tobacco assays, *BAX* was used as the positive control, pGR106 served as the negative control, and buffer was the blank control. At 24 hpi, the same infiltration sites were challenged with BAX at pGR106::*ShROP7*+*BAX* (3) and pGR106+*BAX* (4). Representative leaves were photographed at 5 dpi. Experiments were repeated at least two times with similar results.

### 4.4. Real-Time Quantitative PCR (RT-qPCR) Analysis

Tomato cv. LA1777 and MM seedlings were inoculated with *On*-Lz, and samples were taken at 0, 12, 18, 24, 36, 48, 72, 96, and 120 hpi. Total RNA was extracted according to the bioZOL total RNA extraction kit (Invitrogen, Carlsbad, CA, USA). Reverse transcription of first-strand cDNA was performed according to the manufacturer’s instructions (FastKing RT Kit; Tiangen, Beijing, WI, China). The total reaction system and RT-qPCR program were performed with the UltraSYBR Mixture kit (KangWei, Guangzhou, China). The 2^−∆∆CT^ method was used to calculate relative expression levels. Tomato *GAPDA* was used as a reference gene for expression. Three biological replicates and three technical replicates were performed for each treatment. SPSS 20.0 was used to determine the statistical significance (*p* < 0.05) [42].

### 4.5. TRV2-Mediated Silencing of ShROP7 in Tomato

The tobacco rattle virus (TRV1 and TRV2) was used for the silencing of *ShROP7* in tomato cv. LA1777 and MM. A 248-bp fragment of *ShROP7* was constructed into pTRV2 vector via *BamH*1 digestion and ligation. After the recombinant plasmid was successfully constructed, TRV1 and TRV2-as (anti-sense orientation) were transferred into *Agrobacterium tumefaciens* strain GV3101 respectively, and were cultured and infiltrated as previously described [43]. TRV2::*ShPDS* (phytoene desaturase; sequence accession number NM_001247166) was the positive control. Thirty days after virus inoculation, with the photo-bleaching of positive control plants being observed, the expression level of the *ShROP7* gene was calculated to detect the silencing efficiency.

Once *ShROP7* was silenced, together with the negative control and the *ShROP7*-silenced plants, plants were inoculated with *On*-Lz. The infection phenotypes of tomato leaves were observed and disease index (DI) was evaluated at 7 and 14 dpi. Disease severity was scored according to the following: 0 = no diseased leaves; 1 = 0–5% of leaves having lesions; 3 = 6–10% of leaves having lesions; 5 = 11–20% of leaves having lesions; 7 = 21–40% of leaves having lesions; and 9 = 41–100% of leaves having lesions. The equation of the disease index was calculated as described by Sun et al. [41]. Three biological replicates were performed, with seven leaves evaluated at each timepoint.

For histological observation, *ShROP7*-silenced plants (LA1777) were sampled at 6, 18, 24, 48, and 72 hpi. Samples at each timepoint were stained with 1 mg/mL 3,3-diaminobenzidine (DAB; AMERCO, Solon, OH, USA) for observation of H_2_O_2_ accumulation, and were stained in a 0.05% trypan blue solution to observe the induction of HR cell death. The Olympus BX-51 microscope (Olympus Corp, Tokyo) was used to count the reactive oxygen generation rate (H_2_O_2_ production rate = H_2_O_2_ numbers per 100 penetration sites) and the formation rate of HR cell death (HR production rate = HR numbers per 100 penetration sites). Three to five leaves were observed and at least 50 penetration sites were counted at each timepoint. Standard errors of deviation were calculated using Microsoft Excel. Statistical significance was assessed by a Student’s *t*-test (*P* < 0.01) using SPSS 20.0 software [43]. *ShROP7*-silenced plants (LA1777) were sampled at 0, 24, 48, and 120 hpi for isolating total RNA to analyze the effects of gene expression on plant resistance.

### 4.6. Yeast Two-Hybrid and Bimolecular Fluorescence Complementation (BiFC) Analysis

The full-length sequence of *ShROP7* was cloned into pGBKT7 (BD) and was used as a bait construct. A cDNA library, constructed with tomato cv. LA1777 infected with *On*-Lz, was transformed into the yeast strain Y2H Gold containing pGBKT7-*ShROP7* to screen candidate interaction proteins following the yeast protocols handbook (Takara, Dalian, China). After selection on SD/-Leu-Trp+Aureobasidin (Aba) and SD/-Leu-Trp-His-Ade+Aba+X-α-galactosidase (X-α-Gal) medium, the candidate protein sequences were identified by PCR reaction and were analyzed and compared using the NCBI BLAST tool (Appendix A). The ORF sequence of *ShSOBIR1* without a signal peptide was inserted into pGADT7 (AD) vector. The plasmids of *ShROP7*+*ShSOBIR1* and the negative controls (BD+AD, *ShROP7*+AD, BD+*ShSOBIR1*) were co-transformed into yeast, respectively, and were grown on SD/-Leu-Trp+Aba and SD/-Leu-Trp-His-Ade+Aba+X-α-Gal medium to further verify the interaction of yeast two-hybrid.

The full-length sequence of *ShROP7* and the ORF sequence of *ShSOBIR1* without signal peptide were cloned into pSPYCE and pSPYNE, respectively, for BiFC. The plasmids of *ShROP7*-cYFP and *ShSOBIR1*-nYFP were transformed into *A. tumefaciens* GV3101 for *N. benthamiana* expression. At 48–72 h after Agro-infection, YFP fluorescence was observed using confocal laser scanning microscopy (Olympus FV1000MPE) equipped with a 488 nm filter.

## Figures and Tables

**Figure 1 ijms-23-08557-f001:**
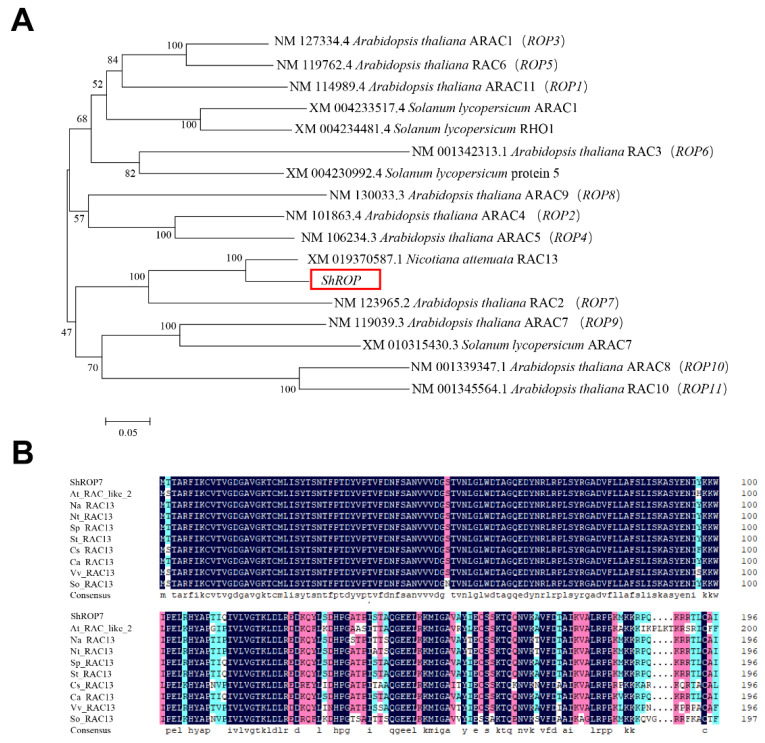
Sequence analysis of Rac-like GTP binding protein ShROP in tomato. (**A**) Phylogenetic analysis of ShROP and other Rac-like GTP binding proteins. MEGA6.0 software was used to analyze phylogeny by proximity method. (**B**) Multiple sequence alignment between ShROP7 and other species RAC13.The amino acid sequence of ShROP7 was compared with *Arabidopsis thaliana* AtRAC-like2 (NP_199409.1), *Nicotiana attenuata* NaRAC13 (XP_019226132.1), *Nicotiana tabacum* NtRAC13 (XP_016451269.1), *Solanum pennellii* SpRAC13 (XP_015065147.1), *Solanum tuberosum* StRAC13 (XP_006367198.1), *Cucumis sativus* CsRAC13 (XP_004134675.1), *Capsicum annuum* CaRAC13 (XP_016562051.1), *Vitis vinifera* VvRAC13 (XP_002277471.1), and *Spinacia oleracea* SoRAC13 (XP_021863578.1).

**Figure 2 ijms-23-08557-f002:**
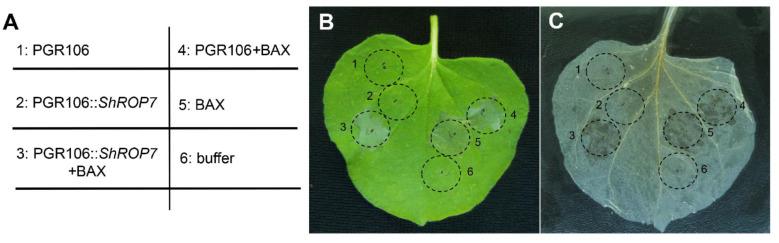
ShROP7 does not induce cell death in tobacco. (**A**) Injection diagram of *Agrobacterium* infected tobacco leaves. (**B**) Typical phenotype of tobacco leaves injected with *Agrobacterium* were photographed at 5 days post-infection. (**C**) The same leaf was decolorized with absolute ethanol and glacial acetic acid (1:1). 1: PGR106, 2: PGR106::*ShROP7*, 3: PGR106::*ShROP7*+BAX, 4: PGR106+BAX, 5: BAX, 6: buffer.

**Figure 3 ijms-23-08557-f003:**
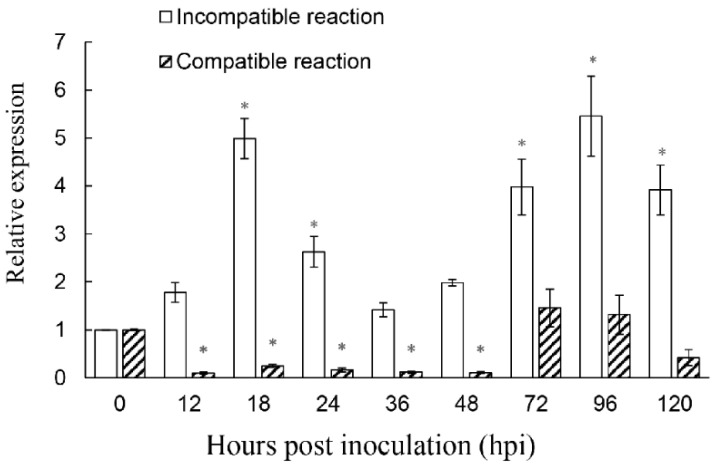
The accumulation of *ShROP7* mRNA is differentially induced in tomato leaves following inoculation with *Oidium neolycopersici*-LZ (*On*-LZ). Error bars represent the variation from three independent replicates. The asterisk (*****) indicates a significant difference between inoculation timepoints and 0 hpi (*p* < 0.05).

**Figure 4 ijms-23-08557-f004:**
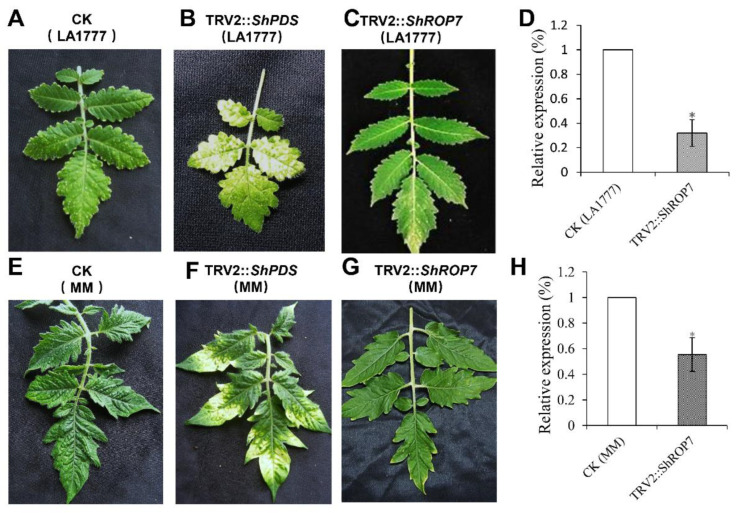
Silencing of *ShROP7* by TRV-VIGS. Phenotypes of control (CK) (**A**), TRV2::*ShPDS* (**B**) and TRV2::*ShROP7* (**C**) tomatoes after 30 days post-infection with *Agrobacterium*. (**D**) Expression level of *ShROP7* in silenced plants (LA1777). Phenotypes of CK (control) (**E**), TRV2::*ShPDS* (**F**) and TRV2::*ShROP7* (**G**) tomatoes after 30 days post infected with *Agrobacterium* in MM plants. (**H**) Expression level of *ShROP7* gene in silenced plants (MM). The asterisk (*) indicate significant differences of disease index between control (CK) and TRV2::*ShROP7* plants (*p* < 0.05).

**Figure 5 ijms-23-08557-f005:**
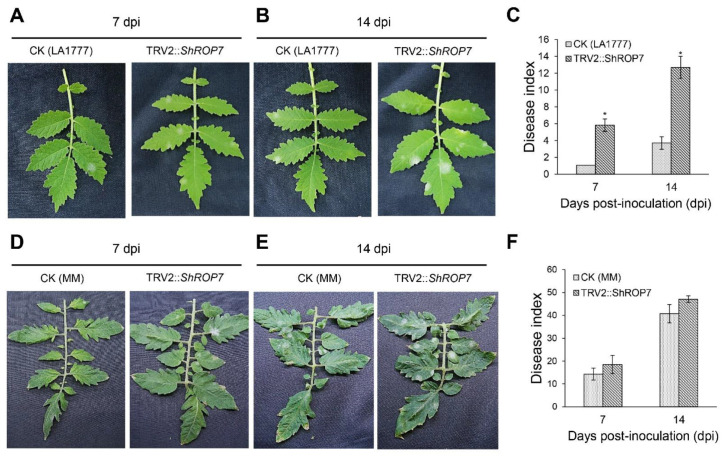
*ShROP7* is required for resistance to powdery mildew. Genetic analysis of resistance response and disease development using TRV-induced gene silencing in tomato. Phenotypes of CK (control) and TRV2::*ShROP7* plants inoculated with *On*-LZ at 7 (**A**) and 14 (**B**) dpi in LA1777 plants and at 7 (**D**) and 14 (**E**) dpi in MM plants. Disease index was calculated for CK and TRV2::*ShROP7* plants at 7 and 14 dpi in LA1777 (**C**) and MM (**F**) plants. The asterisk (*****) indicate significant differences of disease index between control (CK) and TRV2::*ShROP7* plants (*p* < 0.05).

**Figure 6 ijms-23-08557-f006:**
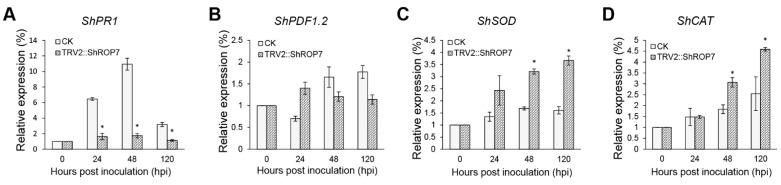
Real-time quantitative PCR (RT-qPCR) analyses of expression of *ShPR1* (**A**), *ShPDF1.2* (**B**), *ShSOD* (**C**) and *ShCAT* (**D**) genes in CK (LA1777) and TRV2::ShROP7 plants inoculated with *On*-LZ. Error bars represent the variations among three independent replicates. The asterisk (*****) indicates a significant difference compared with 0 hpi (*p* < 0.05).

**Figure 7 ijms-23-08557-f007:**
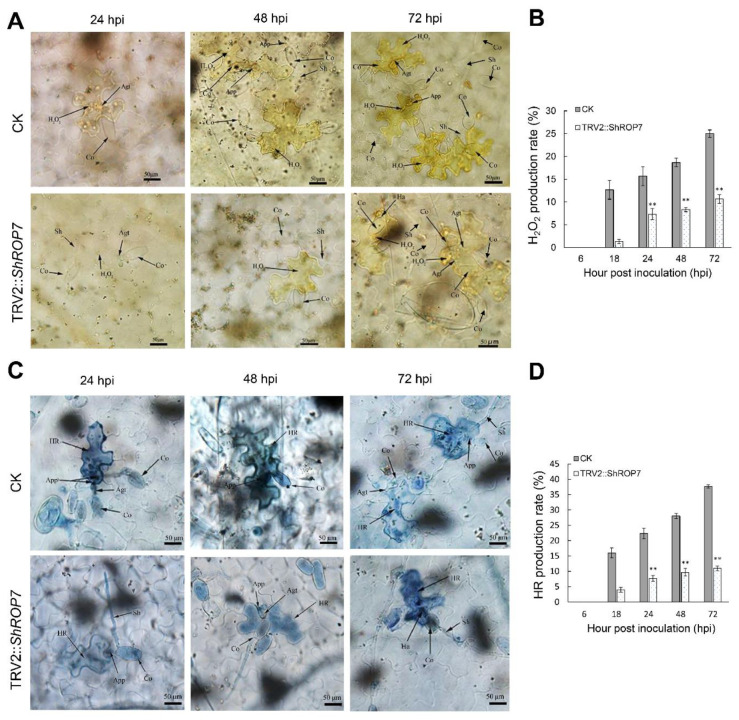
Silencing of *ShROP7* reduced defense responses in tomato following powdery mildew infection. (**A**) Histological observation of H_2_O_2_ accumulation in CK and *ShROP7*-silenced plants inoculated with *On*-LZ at 24, 48 and 72 hpi in LA1777 plants. (**B**) The H_2_O_2_ production rate was calculated at 6, 18, 24, 48 and 72 hpi. (**C**) Histological observation of hypersensitive cell death in TRV2 and *ShROP7*-silenced plants inoculated with *On*-LZ at 24, 48 and 72 hpi in LA1777 plants. Blue (trypan) staining indicates hypersensitive cell death. (**D**) The HR production rate was calculated at 6, 18, 24, 48, and 72 hpi. Error bars represent the variations among three independent replicates. Asterisks (******) indicates statistically significant differences between TRV2 and TRV2::*ShROP7* plants at each time point (*p* < 0.01). Co, conidium; App, appressorium; Agt, appressorium germ tube; Sh, secondary hyphae; Ha, haustorium; Sa, secondary appressorium; HR, hypersensitive response. Bar, 50 μm.

**Figure 8 ijms-23-08557-f008:**
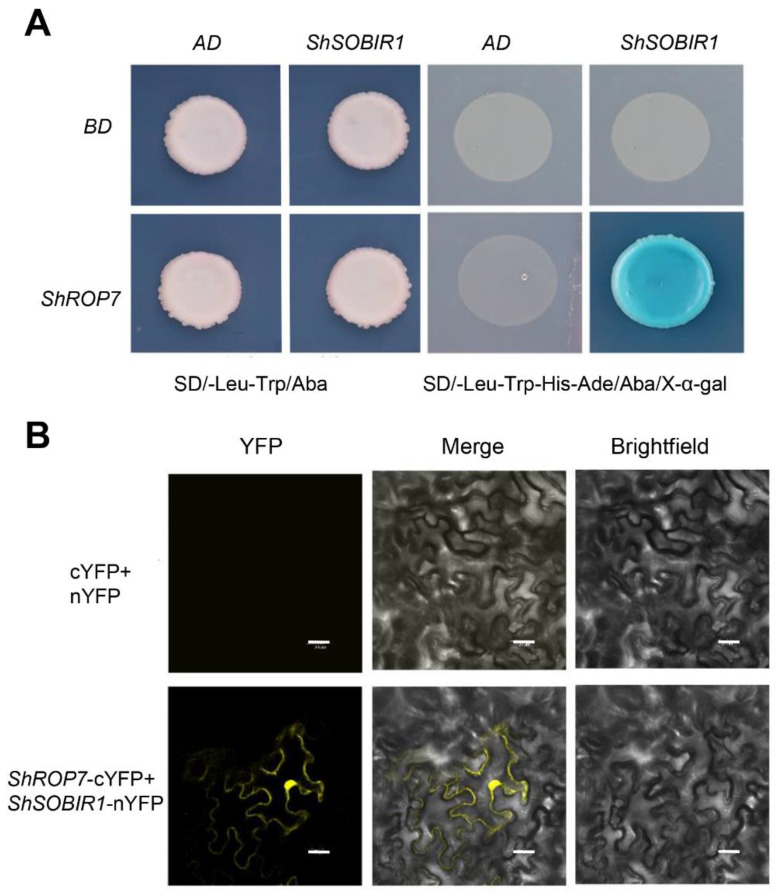
ShROP7 interacts with ShSOBIR1. (**A**) Interaction of ShROP7 and ShSOBIR1 in a yeast two-hybrid system. (**B**) Bimolecular fluorescence complementation (BiFC) analysis of the interaction between ShROP7 and ShSOBIR1. Bars = 20 μm.

**Table 1 ijms-23-08557-t001:** Protein interactions of ShROP7 identified from yeast two-hybrid screen.

Strain No.	Interacting Target Proteins	Accession No.	Description
1	Calcium-binding EF-hand family protein	Solyc03g117470.2.1	Ca2+ binding induces a conformational change in the EF-hand motif, leading to the activation or inactivation of target proteins.
2	60S ribosomal protein L26-1	Solyc02g064670.1.1	Ribosomal large subunit biogenesis; cytoplasmic translation.
3	eukaryotic initiation factor 4A-2	Solyc08g062800.2.1	ATP-dependent RNA helicase activity; RNA secondary structure unwinding; translational initiation.
4	Uncharacterized protein	Solyc05g009220.2.1	Glutamine-tRNA ligase; glutaminyl-tRNA synthetase; belongs to the class-I aminoacyl-tRNA synthetase family.
5	40S ribosomal protein S20-2	Solyc01g096580.2.1	Structural constituent of ribosome; cytoplasmic translation.
6	PGH1	Solyc09g009020.2.1	Magnesium ion binding; phosphopyruvate hydratase activity; glycolytic process.
7	Uncharacterized protein	Solyc07g007680.2.1	Membrane contacts sites (MCSs) domains are exclusively found at MCSs between different organelles; mediates lipid transfer between the two adjacent bilayers independently of membrane fusion and fission reactions.
8	proteasome subunit alpha type-3	Solyc10g081130.1.1	Proteasome subunit alpha type; the proteasome is a multicatalytic proteinase complex.
9	ranslocase of chloroplast34	Solyc03g095220.2.1	GTPase involved in protein precursor import into chloroplasts. Seems to recognize chloroplast-destined precursor proteins and regulate their presentation to the translocation channel through GTP hydrolysis.
10	U3 small nucleolar ribonucleoprotein	Solyc01g007250.2.1	Some proteins in this family suggest a role in ribosome biogenesis and rRNA binding.
11	ERAD-associated E3 ubiquitin-protein ligase component HRD3A	Solyc03g118670.2.1	It mediates protein-protein interactions and the assembly of multiprotein complexes.
12	wound-induced proteinase inhibitor 1	Solyc09g084440.2.1	The proteinase inhibitors inhibit peptidases of the S1 and S8 families; synthesis of the inhibitors throughout the plant is also induced by leaf damage.
13	exonuclease family protein	Solyc01g096570.2.1	Ribonuclease T is responsible for the end-turnover of tRNA and removes the terminal AMP residue from uncharged tRNA.
14	Bap31	Solyc02g032930.2.1	Play a role in endoplasmic reticulum (ER) quality control and sorting
15	Fructose-bisphosphate aldolase	Solyc05g008600.2.1	It is a glycolytic enzyme that catalyzes the reversible aldol cleavage or condensation of fructose-1,6-bisphosphate into dihydroxyacetone- phosphate and glyceraldehyde 3-phosphate.
16	Putative PTI1-like tyrosine-protein kinase 3	Solyc12g098820.1.1	Tyrosine-protein kinases can transfer a phosphate group from ATP to a tyrosine residue in a protein.
17	5-oxoprolinase	Solyc09g010560.1.1	Participate in glutathione metabolism.
18	EIL1	Solyc06g073720.1.1	DNA-binding transcription factor activity; cellular response to iron ion; ethylene-activated signaling pathway.
19	OxaA/YidC-like membrane insertion protein	Solyc05g014050.2.1	Required for the insertion of some light harvesting chlorophyll-binding proteins (LHCP) into the chloroplast thylakoid membrane.
20	Ferredoxin	Solyc10g075160.1.1	Ferredoxins are iron-sulfur proteins that transfer electrons in a wide variety of metabolic reactions.
21	Thioredoxin	Solyc06g060290.2.1	Thioredoxin serves as a general protein disulphide oxidoreductase. It interacts with a broad range of proteins by a redox mechanism based on reversible oxidation of two cysteine thiol groups to a disulphide.
22	Cellulose synthase	Solyc01g087210.2.1	This protein is involved in the pathway plant cellulose biosynthesis.
23	Pectin acetylesterase (PAEs)	Solyc08g005800.2.1	PAEs catalyze the deacetylation of pectin, a major compound of primary cell walls.
24	60S ribosomal protein L24	Solyc09g008800.2.1	L24e/L24 appears to play a role in the kinetics of peptide synthesis and may be involved in interactions between the large and small subunits, either directly or through other factors.
25	Glutathione peroxidase (GSHPx)	Solyc08g006720.2.1	GSHPx is an enzyme that catalyzes the reduction of hydroxyperoxides by glutathione. Its main function is to protect against the damaging effect of endogenously formed hydroxyperoxides.
26	15.7 kDa heat shock protein	Solyc04g014480.2.1	These seem to act as chaperones that can protect other proteins against heat-induced denaturation and aggregation.
27	vesicle transport v-SNARE 12	Solyc05g013050.2.1	SNAP receptor activity; vesicle-mediated transport; intracellular protein transport; V-SNARE proteins are required for protein traffic between eukaryotic organelles.
28	Serinethreonine-protein kinase sty46	Solyc10g055720.1.1	Protein kinases catalyze the transfer of the gamma phosphate from nucleotide triphosphates (often ATP) to one or more amino acid residues in a protein substrate side chain, resulting in a conformational change affecting protein function.
29	DAR GTPase 3	Solyc06g084270.2.1	GTP binding; GTPase activity
30	MAPK16	Solyc12g040680.1.1	Eukaryotic serine-threonine mitogen-activated protein (MAP) kinases are key regulators of cellular signal transduction systems and are conserved; MAPKs play important roles in the signaling of most plant hormones and in developmental processes
31	F-box protein At5g46170	Solyc08g048430.2.1	The F-box is a conserved domain that is present in numerous proteins with a bipartite structure. Through the F-box, these proteins are linked to the Skp1 protein and the core of SCFs (Skp1-cullin-F-box protein ligase) complexes. SCFs complexes constitute a new class of E3 ligases.
32	D111/G-patch domain-containing protein	Solyc08g081580.2.1	It might be a previously undetected RNA-binding domain mediating a distinct type of RNA-protein interaction.
33	Uncharacterized protein	Solyc08g080110.2.1	It is deubiquitinating (DUB) enzymes known as the MINDY family (MIU-containing novel DUB).
34	CHI3	Solyc02g082920.3	Defense against chitinase activity; cell wall macromolecule catabolic process; chitin catabolic process; defense response; polysaccharide catabolic process.
35	GATA transcription factor 5	Solyc01g110310.2.1	Transcriptional activator that specifically binds 5′-GATA-3′ or 5′-GAT-3′ motifs within gene promoters; positive regulation of transcription.

## Data Availability

Data supporting the reported results can be found in the Appendix A.

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
