# Peer review of "Functional Characterization of Tomato ShROP7 in Regulating Resistance against Oidium neolycopersici"

_ijms, 2022, doi:10.3390/ijms23158557_

Round 1

Reviewer 1 Report

In this article "functional characterization of tomato ShROP7 in regulating resistance against Oidium neolycopersici" the authors employed the homology-based cloning technique to clone one of the Rac-like GTP of the tomato and investigate its functional relationship to the powdery mildew infestation. Based on multiple sequence alignment and phylogeny analysis, the authors showed that ShROP7 is close to SlRac13 and Arabidopsis ROP7. Using the quantitative RT-PCR, they further presented that ShROP7 expression significantly increases incompatible interaction but drastically decreases with compatible interaction with the powdery mildew. Furthermore, by silencing the gene using the VIGS technique, the authors presented that the ShROP7 is essential for the resistance to powdery mildew. Finally, using the Y2H system identified several potential interactors and independently confirmed one of these interactions using the BiFC.  

This study is undoubtedly helpful and informative for the reader and helps the scientific community further advance their research in plant-microbe interactions.

However, I find some major and minor issues in the manuscript that certainly need to be addressed for publication.

1)      Please provide appropriate references in lines 29 and 39.

2)      Although the authors mentioned in the text of the result section the schematic diagram of the protein and its different domains, it is not presented in figure no 1.

3)      Also, the descriptions at the beginning of the result section do not align with the figures provided. Please correct these.

4)      For the multiple sequence alignment, surprisingly, the authors did not use the closed homolog of the gene—SlRAC13. It would be nice to include it.

5)      In lines 11-113, the statement seems to be an anonymous reviewer's comment on the manuscript. Therefore, please exclude it from the main text.

6)      The figure legend of Fig.1 does not match the figure provided in the manuscript.

7)      The authors did not explain why they tested ShRO7 for the cell death and necrosis experiment. It would be nice if the authors provided a brief rationale for each experiment that they presented in the result section.

8)      The result section describing figure 3 is confusing. It needs to be rearranged to describe the over-accumulation or reduction of the transcript based on the tomato species tested. They skipped discussing the expression of the gene for each tomato variety.

9)       Also, the authors did not rationalize why they noticed an increase in gene expression for incompatible interactions after a dip.

10)   The authors need to let the reader know why they picked SOD and CAT genes and what is the rationale for testing their transcript level, like the way they did for PR1 and PDF1.2, even though it is evident to the informed readers.

11)   In line 244, the authors mentioned they confirmed the interaction of the candidates identified using Y2H. It would be nice if they provided the method by which they confirmed the interactions of the mentioned candidates.

12)   In line 254, the authors suggested that they did not include the signal peptide region in their Y2H and BiFC study, but they never provided the region of the signal peptide that is excluded in the text.

13)    The BiFC experiment lacks the proper negative control, and the image provided is not that convincing. Please include the appropriate negative and positive controls and provide better images of the interactions.

14)   The discussion section reads more like an introduction. Therefore, please modify it to make it more relevant to your result and discuss the results in detail.

15)   Some of the material and methods sections lack detailed information. Such as 1) method of inoculation of the powdery mildew and at what stage the plants inoculated, 2) Light conditions and light period of the tomato plants growing, 3) parameters used for the software to generate the multiple sequence alignment, predicting the secondary structure of the protein, and building phylogeny tree, 4) parameters of the microscope used to visualize the images for BiFC and others.

Reviewer 2 Report

Thank you very much for the possibility of rewiev the manuscrypt " Functional characterization of tomato ShROP7 in regulating re3 sistance against Oidium neolycopersici ". 

Work is well written, interesting and supported by nice figures. It requires only minor corrections, such as standardizing the size of fonts, asterisks on charts and in materials and methods -abbreviations used for the first time, such as YPDA or HR should be explained.
